# Complexin inhibits spontaneous release and synchronizes Ca²⁺-triggered synaptic vesicle fusion by distinct mechanisms

**Ying Lai**[1,2,3,4†], **Jiajie Diao**[1,2,3,4,5†], **Daniel J Cipriano**[1,2,3,4,5], **Yunxiang Zhang**[1,2,3,4], **Richard A Pfuetzner**[1,2,3,4,5], **Mark S Padolina**[1,2,3,4,5], **Axel T Brunger**[1,2,3,4,5*]

[1]Department of Molecular and Cellular Physiology, Stanford University, Stanford, United States; [2]Department of Neurology and Neurological Science, Stanford University, Stanford, United States; [3]Department of Structural Biology, Stanford University, Stanford, United States; [4]Department of Photon Science, Stanford University, Stanford, United States; [5]Howard Hughes Medical Institute, Stanford University, Stanford, United States

**Abstract** Previously we showed that fast Ca²⁺-triggered vesicle fusion with reconstituted neuronal SNAREs and synaptotagmin-1 begins from an initial hemifusion-free membrane point contact, rather than a hemifusion diaphragm, using a single vesicle–vesicle lipid/content mixing assay (*Diao et al., 2012*). When complexin-1 was included, a more pronounced Ca²⁺-triggered fusion burst was observed, effectively synchronizing the process. Here we show that complexin-1 also reduces spontaneous fusion in the same assay. Moreover, distinct effects of several complexin-1 truncation mutants on spontaneous and Ca²⁺-triggered fusion closely mimic those observed in neuronal cultures. The very N-terminal domain is essential for synchronization of Ca²⁺-triggered fusion, but not for suppression of spontaneous fusion, whereas the opposite is true for the C-terminal domain. By systematically varying the complexin-1 concentration, we observed differences in titration behavior for spontaneous and Ca²⁺-triggered fusion. Taken together, complexin-1 utilizes distinct mechanisms for synchronization of Ca²⁺-triggered fusion and inhibition of spontaneous fusion.

**\*For correspondence:** brunger@stanford.edu

†These authors contributed equally to this work

**Reviewing editor**: Richard Aldrich, The University of Texas at Austin, United States

## Introduction

Synaptic proteins orchestrate Ca²⁺-triggered membrane fusion between synaptic vesicles and the plasma membrane, leading to neurotransmitter release upon an action potential. Key players are neuronal soluble N-ethylmaleimide-sensitive factor attachment protein receptors (SNAREs) synaptobrevin-2/VAMP2 (Vesicle Associated Membrane Protein), syntaxin-1A, and SNAP-25A (Synaptosomal-Associated Protein 25); synaptotagmin; complexin; Munc-18/Sec1, and Munc13 (*Südhof and Rothman, 2009*; *Jahn and Fasshauer, 2012*; *Südhof, 2013*). While it is well established that SNARE proteins provide the minimal energy to drive membrane fusion (*Weber et al., 1998*) and synaptotagmin-1 is the Ca²⁺ sensor for fast synchronous release (*Fernández-Chacón et al., 2001*), the molecular mechanisms of complexin, Munc18, and Munc13 are less certain (*Kümmel et al., 2011*; *Li et al., 2011*; *Ma et al., 2013*; *Trimbuch et al., 2014*). Moreover, the molecular events and protein conformational changes that accomplish Ca²⁺-regulated, fast membrane fusion are still unknown. In this Research Advance we focus on the mechanisms of complexin.

Complexin is a small soluble protein of 134 residues found mainly in the presynaptic terminal. Studies in cortical neuronal cultures revealed that complexin 'activates' fast synchronous release and inhibits spontaneous 'mini' release (*Maximov et al., 2009*; *Kaeser-Woo et al., 2012*); these studies

used RNA-interference double-knockdown of both complexin-1 and complexin-2 isoforms combined with rescue by various truncation and point mutants of complexin-1. In contrast, complexin double-knockout mice showed impairment of both synchronous and spontaneous release in hippocampal neurons (*Reim et al., 2001*; *Xue et al., 2010*). At variance, a complexin double-knockout in *Drosophila* impaired synchronous release but enhanced spontaneous release (*Xue et al., 2009*; *Jorquera et al., 2012*). Overexpression of complexin-1 in PC12 cells suppressed both K$^+$-dependent acetylcholine release (*Itakura et al., 1999*) and evoked release in chromaffin cells (*Archer et al., 2002*). Moreover, expressing synaptobrevin-fused complexin-1 in cultured cortical neurons substantially diminished both spontaneous and evoked neurotransmitter release (*Tang et al., 2006*). Taken together, a consensus has emerged that the two main physiological functions of complexin in neurons at *normal* expression levels consist of activation of synchronous release and suppression of spontaneous release.

Complexin has four domains that are involved in the two main physiological functions of complexin in neurons. The N-terminal domain (residues 1–27) of complexin-1 activates fast synchronous release whereas the accessory helix domain (residues 28–48) is critical for suppressing spontaneous release (*Maximov et al., 2009*). The α-helical SNARE binding domain of complexin-1 (residues 49–70) is essential for all physiological functions; it binds to the groove between the synaptobrevin-2 and syntaxin-1A α-helices in the core part of the neuronal SNARE complex (*Chen et al., 2002*). The C-terminal domain (residues 71–134) has a role in suppressing spontaneous release, but it is expendable for evoked release (*Kaeser-Woo et al., 2012*). The C-terminal domain also promotes vesicle priming (*Kaeser-Woo et al., 2012*) and binding to anionic membranes (*Diao et al., 2013b*).

Previously, we used a single vesicle–vesicle lipid/content mixing assay with reconstituted neuronal SNAREs, synaptotagmin-1, and complexin-1 to decipher the membrane states upon Ca$^{2+}$-triggering, and we found that the presence of complexin-1 greatly enhances the burst of fusion events upon Ca$^{2+}$-injection, that is, complexin-1 effectively synchronizes Ca$^{2+}$-triggered fusion in our minimal system (*Diao et al., 2012*).

## Results

In this Research Advance, we extended our single vesicle–vesicle content mixing assay (*Diao et al., 2012*; *Kyoung et al., 2013*) by monitoring both spontaneous and Ca$^{2+}$-triggered fusion during the same observation cascade (*Figure 1* and 'Materials and methods'). Two types of vesicles were reconstituted: vesicles that mimic the plasma membrane (referred to as *PM vesicles*) and vesicles that mimic synaptic vesicles (referred to as *SV vesicles*) (*Figure 1A,B*).

As before, we observed that complexin-1 synchronizes Ca$^{2+}$-triggered fusion together with synaptotagmin-1 and neuronal SNAREs, producing a higher amplitude upon Ca$^{2+}$ triggering (the probability of fusion per associated vesicle in the first 1-sec time bin upon Ca$^{2+}$-injection with and without complexin-1 is 1.7% and 0.6%, respectively) and a faster decay rate (with complexin-1: 0.43 s$^{-1}$, without complexin-1: 0.11 s$^{-1}$) (*Figure 2A*), in qualitative agreement with studies of cultured cortical mouse neurons and in *Drosophila* (*Reim et al., 2001*; *Maximov et al., 2009*; *Xue et al., 2010*; *Jorquera et al., 2012*). With our improved assay, we also observed that complexin-1 suppresses spontaneous fusion (*Figure 2B*) (the probability of spontaneous fusion per associated vesicle with and without complexin-1 is 0.02% and 0.1% per sec, respectively), in qualitative agreement with knockdown studies in cortical neuronal cultures (*Maximov et al., 2009*), as well as knockout studies in *Drosophila* (*Jorquera et al., 2012*). Thus, in the presence of complexin-1, the ratio of the probability of Ca$^{2+}$-triggered fusion is 85 times higher than the probability of spontaneous fusion, whereas in the absence of complexin this ratio is only 6; so complexin increases this ratio more than ten fold.

To investigate the role of each domain of complexin-1 in synchronizing Ca$^{2+}$-triggered fusion and suppressing spontaneous fusion, we monitored the effects of mutations/truncations of the four domains of complexin-1 that have been studied under physiological conditions (*Maximov et al., 2009*; *Kaeser-Woo et al., 2012*; *Figure 2C*). The probability of spontaneous fusion was determined by counting the number of associated SV vesicles that underwent fusion during the spontaneous fusion periods during multiple repeat experiments; the higher the number in the bar chart, the more spontaneous fusion events occurred (*Figure 2E* and *Figure 2—source data 1*). To assess the degree of synchronization of Ca$^{2+}$-triggered fusion events we determined the rate of the decay of the respective fusion histograms (*Figure 2F* and *Figure 2—source data 1*) and also plotted the amplitude of the histograms upon Ca$^{2+}$-injection (*Figure 2G*); the higher the numbers in these respective bar charts, the more synchronization occurs. For comparison of these quantities, we normalized panels E and G, and used the same concentration for wild type and the complexin-1 mutants (2 µM).

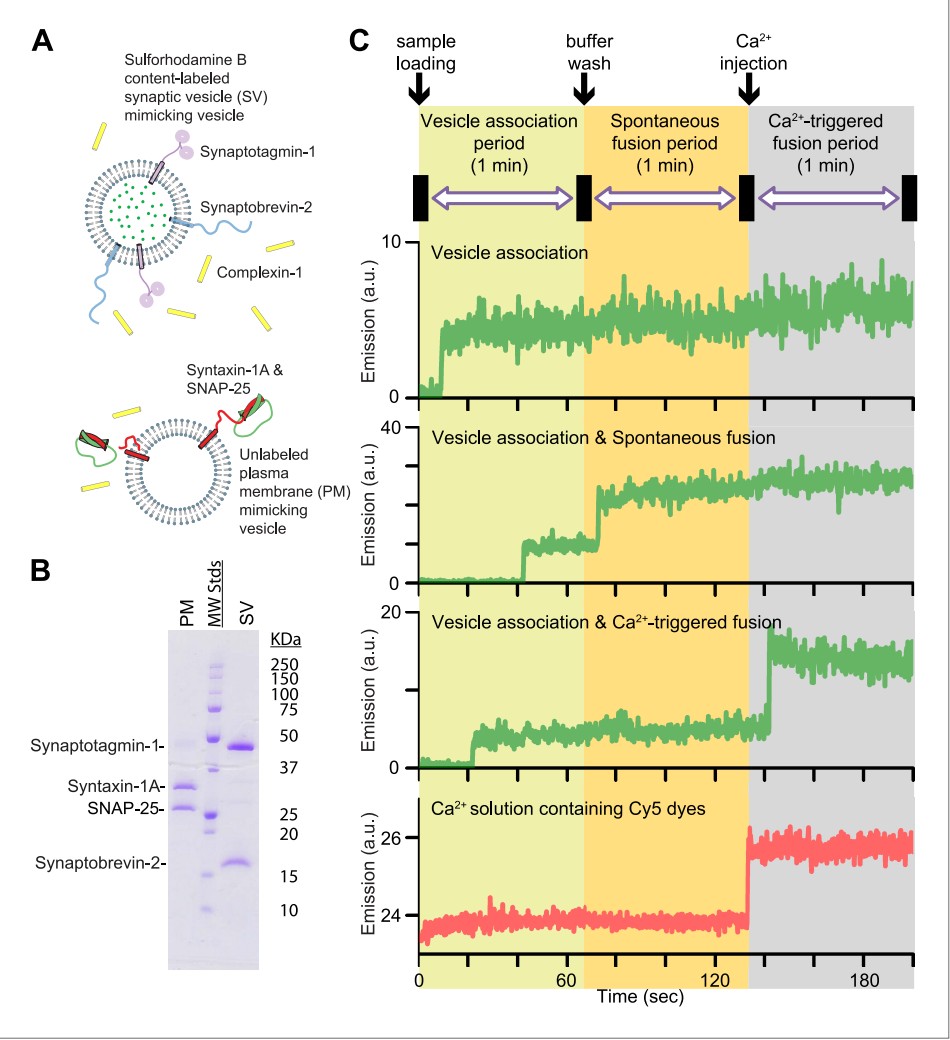

**Figure 1**. Reconstituted single vesicle-vesicle fusion assay. (**A**) Protein composition and labeling of proteoliposomes. (**B**) SDS-PAGE analysis of reconstituted vesicles mimicking the plasma membrane (PM) and synaptic vesicles (SV). (**C**) Schema of the extended single vesicle–vesicle content mixing assay. The shaded backgrounds indicate the three subsequent 1 min time periods of the procedure and the short intervals (5 s) for buffer exchanges. Representative content fluorescence intensity time traces are shown for an associated pair of SV and PM vesicles without any fusion (top trace), for an associated pair that undergoes spontaneous fusion, and for an associated pair that undergoes fusion after $Ca^{2+}$-injection. The association of an SV vesicle to a surface-immobilized PM vesicle is characterized by fluorescence intensity increases during the 1 min period after initial SV vesicle loading. Spontaneous and $Ca^{2+}$-triggered fusion events are characterized by a subsequent stepwise increase of the content dye fluorescence intensity during the respective 1 min observation periods. The time point of the arrival of $Ca^{2+}$ is determined by the appearance of fluorescence intensity from soluble Cy5 dyes that are part of the injected solution (bottom trace).

We found that the central SNARE-binding domain of complexin-1 is essential for both spontaneous and $Ca^{2+}$-triggered fusion since mutation of four residues in that domain (R48A, R59A, K69A, Y70A, C105A, termed '4M' mutant [***Maximov et al., 2009***]) is similar to the control without complexin-1 (compare first and last columns in ***Figure 2D–G***). The N-terminal domain is essential for synchronization of $Ca^{2+}$-triggered fusion, but not for the suppression of spontaneous fusion (third columns in ***Figure 2D–G***). The opposite is true for the C-terminal domain: it is essential for suppressing spontaneous fusion, but not for synchronization (fifth columns in ***Figure 2D–G***). Moreover, the accessory domain is essential for the suppression of spontaneous fusion (fourth columns in ***Figure 2D–G***). All these results are in excellent qualitative agreement with studies performed with cultured cortical neurons (***Maximov et al., 2009***; ***Kaeser-Woo et al., 2012***).

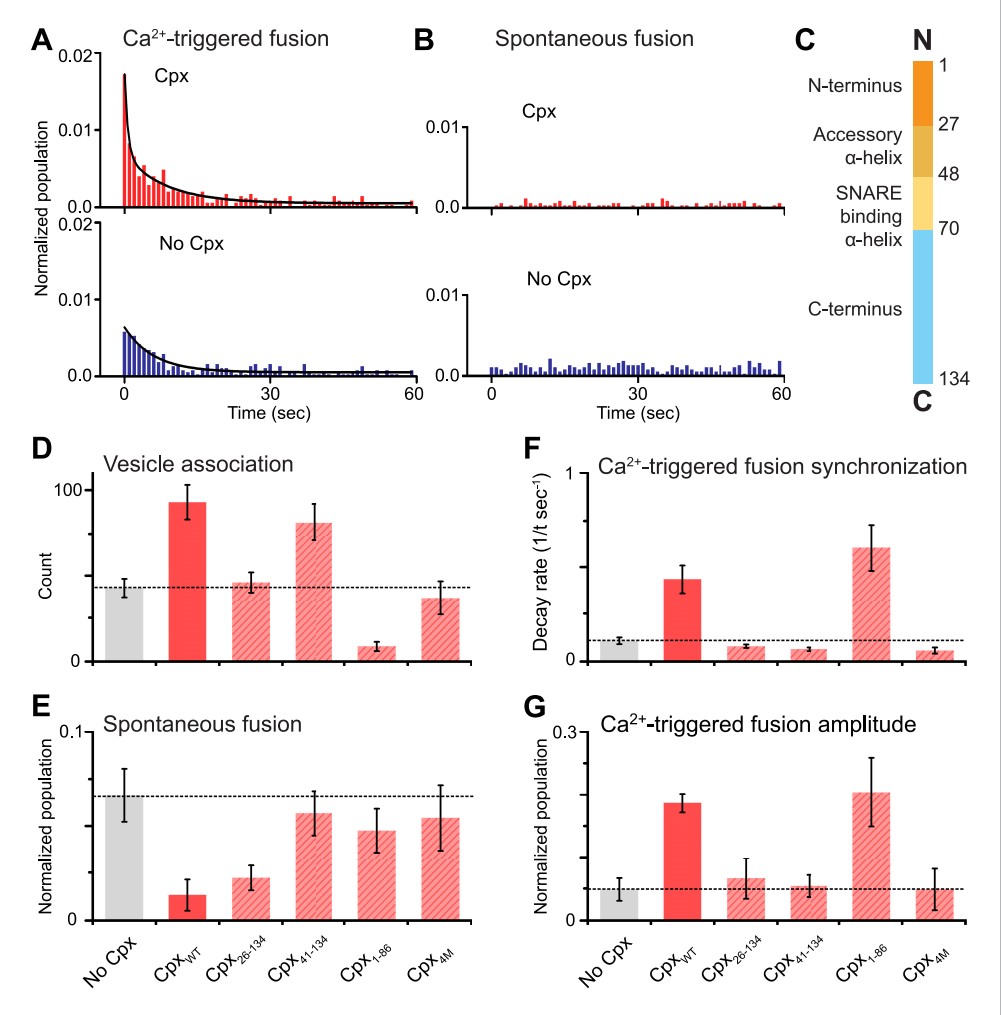

**Figure 2**. Complexin-1 synchronizes Ca²⁺-triggered fusion and inhibits spontaneous fusion in concert with synaptotagmin-1 and neuronal SNAREs. (**A** and **B**) Probability of fusion vs time upon 500 μM Ca²⁺-injection and spontaneous fusion, both in the presence of full-length synaptotagmin-1 and neuronal SNAREs, but with and without 2 μM complexin-1 (Cpx) as indicated. (**A**) The histograms of Ca²⁺-triggered fusion events (1 s time bin) were normalized by the number of associated SV vesicles and fit to an exponential decay function. The decay rate with and without complexin-1 is 0.43 s⁻¹ and 0.11 s⁻¹ respectively. (**B**) The histograms of spontaneous fusion events were normalized by the number of associated SV vesicles. (**C**) Domain structure of complexin-1. To probe the roles of the four domains of complexin-1 we used the truncation mutant Cpx$_{26–134}$ for the N-terminal domain, the truncation mutant Cpx$_{41–134}$ for the accessory α-helical domain, the Cpx$_{4M}$ mutant for the SNARE binding domain, and the truncation mutant Cpx$_{1–88}$ for the C-terminal domain. We chose these particular truncations/mutations based on studies of cortical neuronal cultures (***Maximov et al., 2009***; ***Kaeser-Woo et al., 2012***). (**D–G**) The bar graphs show the effects of complexin-1 and its mutants on SV-PM vesicle association (**D**), the number of spontaneous fusion events over the 1-min observation period divided by the number of associated SV vesicles (**E**), the decay rate of the histogram upon Ca²⁺-injection (**F**), and the amplitude of the first 1-sec time bin upon Ca²⁺-injection (**G**). Each value in panel **G** was normalized by the respective number of fusion events after Ca²⁺-injection. Moreover, only those single vesicle–vesicle fluorescence intensity traces that exhibited a distinct event during the association period were analyzed for fusion events during the subsequent observation periods. Error bars in **D**, **E**, **G** are standard deviations for 6–13 independent repeat experiments. Error bars in **F** are error estimates computed from the covariance matrix upon fitting the corresponding histograms with a single exponential decay function using a Levenberg-Marquardt technique.

The following source data is available for figure 2:

**Source data 1**. Shown are the data that were used to generate panels D-G of *Figure 2*: individual histograms of probability of fusion vs time for the complexin-1 mutants, for spontaneous fusion (upper panels), and upon 500 μM Ca²⁺-injection (lower panels).

We next studied the effect of complexin-1 on spontaneous and Ca²⁺-triggered fusion as a function of complexin-1 concentration ranging from 0–2 µM (*Figure 3* and *Figure 3—source data 1*). The suppression of spontaneous fusion by complexin-1 becomes significant between 100 and 500 nM complexin-1 (*Figure 3B*). In contrast, the effect of complexin-1 on the synchronization of Ca²⁺-triggered fusion becomes significant at lower concentration, between 50 and 100 nM complexin (*Figure 3C,D*). As a control, addition of an excess of the soluble fragment of synaptobrevin-2 (a.a. 1–96) reduced the association between SV and PM vesicles to background levels even at the highest complexin-1 concentration tested (2 µM), indicating that complexin-1 alone does not cause significant SV-PM vesicle association (*Figure 3A*, right-most column).

Finally, we investigated if synaptotagmin-1 plays a role in suppressing spontaneous release. As before, we prepared SV vesicles with both reconstituted synaptobrevin-2 and synaptotagmin-1. To evaluate the

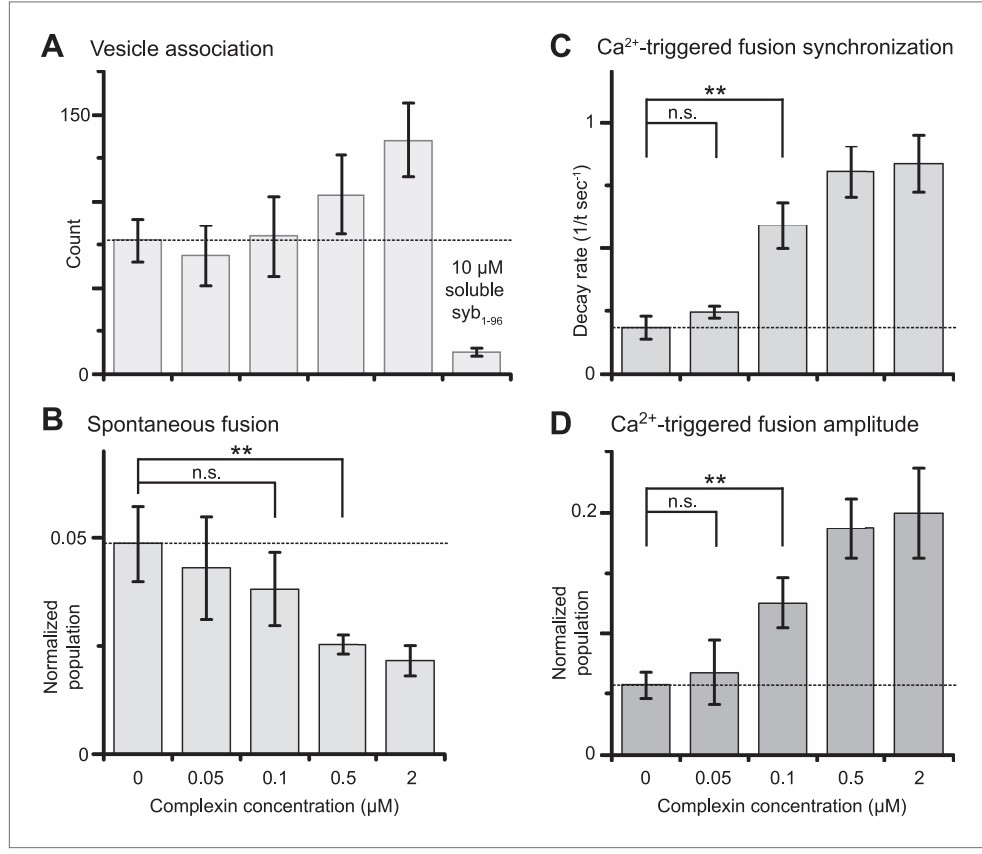

**Figure 3**. Suppression of spontaneous fusion and synchronization of Ca²⁺-triggered fusion vs complexin-1 concentration (in concert with synaptotagmin-1 and neuronal SNAREs). The bar graphs show the effect of wildtype complexin-1 at specified concentrations on SV-PM vesicle association (**A**), the number of spontaneous fusion events divided by the number of associated SV vesicles (**B**), the decay rate of the histogram of fusion events upon Ca²⁺-injection (**C**), and the amplitude of the first 1-sec time bin upon Ca²⁺-injection (**D**). Each value in panel **D** was normalized by the respective number of fusion events after Ca²⁺-injection. Moreover, only those single vesicle–vesicle fluorescence intensity traces that exhibited a distinct event during the association period were analyzed for fusion events during the subsequent observation periods. Error bars in **A**, **B**, **D** show the standard deviation for 6–10 independent repeat experiments. Error bars in **C** are error estimates computed from the covariance matrix upon fitting the corresponding histograms with a single exponential decay function using a Levenberg-Marquardt technique. ** indicates $p<0.01$ using the Student's *t* test.

The following source data is available for figure 3:

**Source data 1**. Shown are the data that were used to generate the bar charts in *Figure 3*: individual histograms of probability of fusion vs time at specified complexin-1 concentrations, for spontaneous fusion (upper panels) and upon 500 µM Ca²⁺-injection (lower panels).

role of synaptotagmin-1 we also prepared vesicles with just reconstituted synaptobrevin-2. We found that the suppression of spontaneous fusion depends only on the presence of complexin-1, but not synaptotagmin-1 (*Figure 4* and *Figure 4—source data 1*). However, both complexin-1 and synaptotagmin-1 contribute to SV-PM vesicle association (in conjunction with neuronal SNAREs) in an approximately additive fashion.

## Discussion

Our improved single vesicle–vesicle content-mixing assay mimics characteristics of $Ca^{2+}$-triggered synaptic vesicle fusion as well as spontaneous release. Our assay qualitatively reproduces the effects of complexin-1 and key mutations on both spontaneous release and evoked release that have been observed in cortical neuronal cultures (*Maximov et al., 2009*; *Kaeser-Woo et al., 2012*; *Figure 2*).

Previous reconstituted assays of complexin function led to some apparently conflicting conclusions. For example, complexin-1 suppressed SNARE-mediated content-mixing in a cell-based assay, and the combination of synaptotagmin-1 and $Ca^{2+}$ reversed this effect, albeit on a very slow time scale (*Giraudo et al., 2009*). Complexin-1 promoted both spontaneous and $Ca^{2+}$-triggered lipid mixing using a single vesicle–vesicle lipid-mixing assay together with neuronal SNAREs, but, surprisingly, in the absence of synaptotagmin-1 (*Yoon et al., 2008*). In contrast, complexin-1, together with neuronal SNAREs and synaptotagmin-1, reduced $Ca^{2+}$-independent lipid-mixing and it enhanced $Ca^{2+}$-triggered lipid mixing using an ensemble assay (*Malsam et al., 2012*). Moreover, this ensemble fluorescence study revealed enhanced lipid-mixing upon $Ca^{2+}$ triggering with a complexin-1 fragment without the N-terminal and accessory domains. Thus, this lipid-mixing result does not correlate with the observation that the N-terminus of complexin-1 is essential for activation of fast synchronous release (*Figure 1E* in *Maximov et al., 2009*). In contrast, our single vesicle–vesicle content-mixing assay correlates well with the physiological data since the N-terminal and accessory domains are required for synchronization of $Ca^{2+}$-triggered fusion (*Figure 2F,G*). In retrospect, many previous studies suffered from the lack of a genuine content mixing indicator and the use of an ensemble method that cannot distinguish between effects due to vesicle association and fusion, see discussion in refs. (*Diao et al., 2012*, *2013c*). Our single vesicle–vesicle content mixing assay addresses both limitations. Another single vesicle–vesicle content mixing assay was also developed using a much larger probe (dual-labeled DNA hairpin) (*Diao et al., 2010*); using this assay, accelerated fusion pore expansion by complexin-1 was observed (*Lai et al., 2013*).

Our single vesicle–vesicle content mixing assay revealed that the effect of complexin-1 on synchronization of $Ca^{2+}$-triggered fusion becomes significant between 50 and 100 nM, well above the $k_D$ of the interaction between complexin-1 and the ternary SNARE complex of ~10 nM (*Xu et al., 2013*), yet the effect on spontaneous fusion becomes only significant at higher complexin concentrations (*Figure 3*). Models of complexin function have primarily focused on a clamping or hindrance mechanism that restricts spontaneous release (*Giraudo et al., 2006*; *Krishnakumar et al., 2011*; *Kümmel et al., 2011*; *Li et al., 2011*; *Trimbuch et al., 2014*), along with a model that combined the clamping role with the facilitation/synchronization role of complexin-1 and, in particular, its N-terminal domain (*Xue et al., 2010*). However, if the same mechanism would facilitate synchronization as well, one would expect identical behaviors as the concentration of complexin-1 is increased. Yet, the different titration behaviors (*Figure 3*) suggest that there are differences in mechanisms for synchronization of $Ca^{2+}$-triggered fusion and for suppressing spontaneous fusion. Moreover, genetic manipulations of the single complexin homolog in *Drosophila* show that the suppression and synchronization functions are genetically separable, also supporting the notion of distinct mechanisms in *Drosophila* (*Cho et al., 2014*).

Complexin-1 and binary (syntaxin-1A/SNAP-25) SNARE complex interact weakly, as measured by a pull-down assay (*McMahon et al., 1995*) and by single molecule experiments (*Weninger et al., 2008*), although studies by ITC and fluorescence anisotropy did not reveal a significant interaction (*Pabst et al., 2002*). The dissociation constant of an interaction between the accessory helical domain and a partially assembled SNARE complex had been measured to be in the low µM range (*Kümmel et al., 2011*) although a re-examination of this interaction has cast doubt on the original derivation of that estimate (*Trimbuch et al., 2014*). Finally, the C-terminus of complexin-1 interacts with anionic membranes, but no estimate for the affinity is available (*Diao et al., 2013b*). In any case, the C-terminus is not essential for the synchronization function of complexin-1 (*Figure 2*). It is of course possible that the weak interactions between complexin-1 and the binary complex or partially folded SNARE complex and the membrane become more significant in the context of molecular crowding at the membrane point contact.

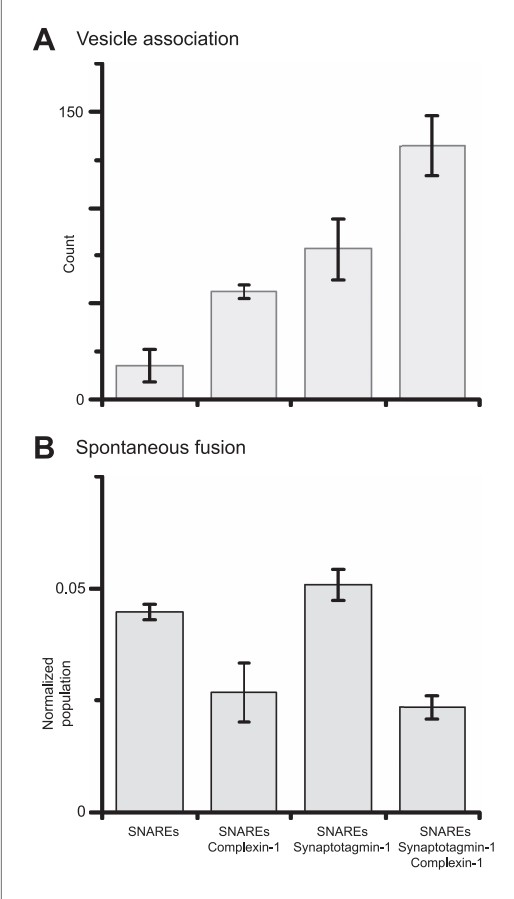

**Figure 4**. Suppression of spontaneous fusion by complexin-1 with or without synaptotagmin-1. The bar graphs summarize the effect of 2 µM complexin-1 on SV-PM vesicle association (**A**) and the number of spontaneous fusion events divided by the number of associated SV vesicles (**B**); the conditions that did not include synaptotagmin-1 used SV vesicles that were reconstituted with synaptobrevin-2 only. Moreover, only those single vesicle–vesicle fluorescence intensity traces that exhibited a distinct event during the association period were analyzed for spontaneous fusion events.

The following source data is available for figure 4:

**Source data 1**. Shown are the data that were used to generate the bar chart in **Figure 4B**: individual histograms of probability of spontaneous fusion vs. time at specified conditions.

Taken together, we propose that the synchronization function of complexin-1 is primarily driven by the core interaction with the SNARE complex in concert with the N-terminal domain and with possible assistance from the accessory domain. The suppression of spontaneous fusion also requires the core interaction and the accessory domain. However, in contrast to synchronization, suppression of spontaneous fusion involves membrane binding by the C-terminal domain of complexin-1, but the N-terminal domain is not involved. These distinct roles of the N- and C-terminal domains may explain the different titration behaviors (**Figure 3**) and suggest distinct mechanisms for both functions of complexin-1.

## Materials and methods

### Protein purification

#### Synaptobrevin and syntaxin

Full-length, cysteine-free rat synaptobrevin-2 (VAMP2) and rat syntaxin-1A (Stx1a) were prepared using the same protocols. They were expressed separately from plasmid pGEX-KG (**Guan and Dixon, 1991**) with a N-terminal glutathione S-transferase tag proceeded by a thrombin cleavage site to remove the tag, and all cysteines were changed to alanine. The proteins were expressed in different cultures of *Escherichia coli* BL21 (DE3) (Novagen, EMD Chemicals, Gibbstown, NJ) by growing the cells to $OD_{600}$ between 0.6 and 0.8 at 37°C, and then protein expression was induced at 16°C 100 rpm for 12–16 hr with 0.1 mM IPTG. Cell pellets from 1 l of culture were suspended in 40 ml of sodium phosphate pH 8, 1 mM dithiothreitol (DTT), 0.2% Triton X-100, 1% dodecylmaltoside (DDM, Anatrace, Maumee, OH), and 0.2 mM phenylmethylsulfonyl fluoride (PMSF) supplemented with complete protease inhibitor cocktail tablets (Roche, Basel, Switzerland), and the cells were disrupted by sonication at 5/15 s on/off pulse for 5 min at 50% power using a Sonicator Ultrasonic Processor XL-2020 (Misonix, Farmingdale, New York, USA). The extract was centrifuged by a 10 min 15,000 rpm spin in JA-20 (Beckman Coulter, Brea, CA) rotor and further cleared by centrifugation at 40,000 rpm for 35 min in a Ti-45

(Beckman Coulter, Brea, CA) rotor. The supernatant was incubated for 2 hr with a 1.5 ml slurry of Glutathione Sepharose 4 Fast Flow beads (GE Healthcare, Uppsala, Sweden) that was pre-equilibrated in the same buffer. The beads were then collected in a column and washed with 80 column volumes of buffer containing 50 mM sodium phosphate, pH 8, 1 mM DTT, and 0.2% Triton X-100 and then with 30 column volumes of buffer containing 40 mM Tris–HCl, pH 8, 150 mM NaCl, 1 mM DTT, and 0.8% n-octyl-β-D-glucopyranoside (OG, Anatrace, Maumee, OH). The beads were then incubated for 90 min at ambient temperature with 100 units of bovine a-thrombin (Haematologic Technologies, Essex Junction, Vermont, USA) in buffer containing 40 mM Tris–HCl pH 8, 150 mM NaCl, 1 mM DTT,

3% OG, and 10% glycerol. The supernatant containing cleaved protein was flowed off the column after which thrombin activity was quenched with 1 mM PMSF. The concentration of the proteins was measured by absorbance at 280 nm, and aliquots were either reconstituted or frozen in liquid nitrogen. Although the syntaxin-1A construct was designed to include the N-terminus of syntaxin-1A, the application of thrombin proteolysis also removed nine additional residues at the N-terminus as determined by Edman sequencing performed by the Stanford Protein and Nucleic Acid Facility. Removal of these nine N-terminal residues has no effect on fusion efficiency and kinetics for spontaneous and $Ca^{2+}$-triggered fusion with neuronal SNAREs, synaptotagmin-1, and complexin-1 (YZ et al., in preparation).

Soluble rat synaptobrevin-2 (residues 1–96) ($Syb_{1-96}$) was expressed in BL21 (DE3) with an N-terminal TEV cleavable hexa-histidine tag. The protein was purified by Ni-NTA affinity chromatography using standard procedures and buffers (Qiagen, Hilden, Germany), digested overnight with TEV protease, and further purified by size exclusion chromatography using a HiLoad Superdex 200 16/600 column (GE Healthcare, Uppsala, Sweden) that was pre-equilibrated with 25 mM Tris–HCl, pH 8.0, 50 mM NaCl, 0.5 mM EDTA and 0.5 mM TCEP.

## SNAP-25

Cysteine-free SNAP-25 (C84S, C85S, C90S, and C92S) was expressed from plasmid pTEV5 (*Rocco et al., 2008*) with an N-terminal TEV protease cleavable hexa-histidine tag. The proteins were expressed overnight in autoinducing media (*Studier, 2005*) in *E. coli* strain BL21(DE3) at 30°C. Cells from 4 l of culture were resuspended in 200 ml of 50 mM sodium phosphate, pH 8.0, 300 mM NaCl, 20 mM imidazole, and 10% glycerol (wt/vol) supplemented with 1 mM PMSF and 4 EDTA-free protease inhibitor cocktail tablets. Cells were lysed by three passes through the Emulsiflex C5 homogenizer (Avestin, Ottawa, Canada) at 15,000 psi. The lysate was clarified by centrifugation in the Ti-45 rotor for 1.5 hr at 40,000 rpm. The supernatant was bound to a 5 ml Nickel-NTA column by flowing the lysate on to the column at 1 ml/min using an AKTA Prime (GE Healthcare, Uppsala, Sweden). The column was washed with 150 ml of SNAP-25 buffer containing 50 mM sodium phosphate, pH 8.0, 300 mM NaCl, 20 mM imidazole, 10% glycerol (wt/vol), supplemented with 50 mM imidazole and eluted with buffer containing 50 mM sodium phosphate, pH 8.0, 300 mM NaCl, 20 mM imidazole, 10% glycerol (wt/vol) and 350 mM imidazole. Protein containing fractions were combined, DTT was added to 5 mM, EDTA was added to 1 mM, and 150 µg of TEV protease was added to remove the hexa-histidine tag. This mixture was dialyzed against buffer containing 20 mM HEPES, pH 7.5, 100 mM NaCl, 4 mM DTT, and 10% glycerol (wt/vol) overnight at 4°C. The TEV-cleaved SNAP-25 was concentrated in a 15 ml Amicon ultra centrifugal concentrator with a 10,000 molecular weight cutoff dialysis cassette (Millipore, Billerica, MA) to 5 ml and injected on the Superdex 200 (16/60) column (GE Healthcare, Uppsala, Sweden) equilibrated with buffer containing 20 mM HEPES, pH 7.5, 100 mM NaCl, 4 mM DTT, 10% glycerol (wt/vol). Protein containing fractions were combined and then concentrated using a 3000 molecular weight cutoff dialysis cassette (Millipore, Billerica, MA). The concentration of SNAP-25 was measured by absorbance at 280 nm, and aliquots were frozen in liquid nitrogen.

## Synaptotagmin

Full-length rat synaptotagmin-1 was expressed from plasmid pJ414 (DNA 2.0 Menlo Park, CA) with a C-terminal deca-histidine tag proceeded by a PreScission protease cleavage site to remove the tag, and all cysteines were changed to alanine except the cysteine residue at position 277. Synaptotagmin-1 was expressed in *E. coli* BL21 (DE3) by growing the cells to $OD_{600}$ between 0.6 and 0.8 at 37°C then induced at 20°C for 12-16 hr with 0.5 mM IPTG. The cells from 6 l of induced culture were harvested and suspended in 200 ml of buffer containing 50 mM sodium phosphate, pH 7.4, 600 mM NaCl, 2 mM DTT, and 10% glycerol (wt/vol) supplemented with PMSF to 1 mM, and two EDTA free complete protease inhibitor cocktail tablets. Cells were lysed by three passes through the Emulsiflex C5 homogenizer at 15,000 psi. Cell debris and inclusion bodies were removed by centrifugation at 8000 rpm in a Beckman JA-20 rotor (Beckman Coulter, Brea, CA) for 10 min. The supernatant was then centrifuged at 8000 rpm for 10 min in the same rotor. Membranes were collected by centrifugation at 40,000 rpm in a Beckman Ti-45 rotor for 1 hr. Membranes were washed by homogenization in 100 ml of buffer containing 50 mM sodium phosphate, pH 7.4, 600 mM NaCl, 5 mM DTT, 1 mM PMSF, 10% glycerol (wt/vol) supplemented with two EDTA-free complete protease inhibitor cocktail tablets. Membranes were harvested again by centrifugation in a Ti-45 rotor at 40,000 rpm for 1 hr. The pellet was suspended in 100 ml of buffer containing 50 mM sodium phosphate, pH 7.4, 600 mM NaCl, 5 mM DTT,

1 mM PMSF, 10% glycerol (wt/vol) supplemented with 2 EDTA-free complete protease inhibitor cocktail tablets and frozen in two 50 ml aliquots in liquid nitrogen. One aliquot of membranes was solubilized in the presence of 1.5% DDM for 1 hr at 4°C. The extract was clarified by centrifugation in a Ti-45 rotor at 40,000 rpm for 35 min. The supernatant was bound to a 4 ml bed volume of Nickel-NTA beads (Qiagen, Hilden, Germany) equilibrated in buffer containing 50 mM sodium phosphate, pH 7.4, 600 mM NaCl, 2 mM DTT, and 10% glycerol (wt/vol), and 1.5% DDM by stirring at 4°C for 1 hr. Beads were harvested by centrifugation and poured into a column, attached to an AKTA Prime and washed with 50 ml of buffer containing 50 mM sodium phosphate, pH 7.4, 600 mM NaCl, 2 mM DTT, 10% glycerol (wt/vol), and 110 mM OG, and then eluted with the same buffer containing 500 mM imidazole. Protein containing fractions were combined and injected on a Superdex 200 (16/60) column equilibrated in 20 mM sodium phosphate, pH 7.4, 300 mM NaCl, 2 mM DTT, 110 mM OG, and 10% glycerol (wt/vol). Peak fractions were then combined and 100 µg of PreScission protease (GE Healthcare, Uppsala, Sweden) was added to cleave the tag. Cleaved synaptotagmin-1 was diluted with 20 mM sodium phosphate, pH 7.4, 100 mM NaCl, 2 mM DTT, 110 mM OG, 20 µM EGTA, and 10% glycerol (wt/vol) to bring the NaCl to 100 mM then injected on a MonoS 5/50 column (GE Healthcare, Uppsala, Sweden), washed with the same buffer and then eluted with buffer containing 20 mM sodium phosphate, pH 7.4, 600 mM NaCl, 2 mM DTT, 110 mM OG, 20 µM EGTA, and 10% glycerol (wt/vol). Protein containing fractions were dialyzed against 20 mM sodium phosphate, pH 7.4, 300 mM NaCl, 2 mM DTT, 110 mM OG, 10% glycerol (wt/vol) in order to lower the salt concentration and used for reconstitution into proteoliposomes.

## Complexin-1 and the 4M mutant of complexin-1

Full-length wild-type complexin-1 (Cpx) and the 4M mutant of complexin-1 (Cpx$_{4M}$) (R48A, R59A, K69A, Y70A, C105A) were expressed separately in *E. coli* using BL21 (DE3) cells with a thrombin-cleavable N-terminal hexa-histidine tag from plasmid pET28a (Novagen, EMD Chemicals, Gibbstown, NJ). Cells were grown in Luria Broth to an OD$_{600}$ between 0.6 and 0.8 and protein expression was induced by the addition of IPTG to 0.5 mM for 12–16 hr at 20°C. Cells were harvested by centrifugation and resuspended in 20 mM HEPES, pH 7.5, 500 mM NaCl, 2 mM DTT, and 10 mM imidazole supplemented with 1 mM PMSF and EDTA-free complete protease inhibitor cocktail tablets. Cells were lysed by passing them through the Emulsiflex C5 homogenizer at 15,000 psi three times. Lysate was clarified by centrifugation in a Ti-45 rotor for 35 min at 40,000 rpm. Supernatant was bound to a 4 ml bed volume of Nickel-NTA beads in batch, stirring for 1 hr at 4°C. The beads were harvested by centrifugation and poured into a column, attached to an ATKA Prime, and washed with 60 ml of buffer containing 20 mM HEPES, pH 7.5, 500 mM NaCl, 2 mM DTT, 10 mM imidazole, and 25 mM imidazole and then eluted with the same buffer containing 450 mM imidazole. Protein-containing fractions were combined and 100 units of thrombin (Haematologic Technologies, Essex Junction, Vermont, USA) were added. The same was then dialyzed against 500 ml of 20 mM HEPES, pH 7.5, 50 mM NaCl, and 4 mM DTT overnight at 4°C. PMSF was added to 1 mM to quench the thrombin activity and the mixture was injected on to a MonoQ column 5/50 equilibrated with 20 mM HEPES pH 7.5, 50 mM NaCl, 4 mM DTT. The column was washed with 20 column volumes of the same buffer and the protein eluted with a linear NaCl gradient in 20 mM HEPES pH 7.5, 4 mM DTT, and from 50 mM to 500 mM NaCl over 30 column volumes. The protein containing fractions were combined and dialyzed against 20 mM HEPES, pH 7.5, 100 mM NaCl, 4 mM DTT overnight at 4°C and then concentrated using a 3000 molecular weight cutoff dialysis cassette (Millipore, Billerica, MA). The protein concentration was measured by absorbance at 280 nm, and aliquots were frozen in liquid nitrogen.

## Complexin-1 truncation mutants

Complexin-1 26–134 (Cpx$_{26-134}$), complexin-1 41–134 (Cpx$_{41-134}$), and complexin-1 1–86 (Cpx$_{1-86}$) were expressed in *E. coli* as N-terminal GST fusions from plasmid pGEX-KT. BL21 (DE3) cells (Novagen, EMD Chemicals, Gibbstown, NJ) transformed with the appropriate plasmid DNA were autoinduced overnight at 37°C (**Studier, 2005**). Induced cells from 4 l of culture were each harvested and resuspended separately in 200 ml of 10 mM Na$_2$HPO$_4$, pH 7.4, 2 mM KH$_2$PO$_4$, 137 mM NaCl, 2.7 mM KCl (PBS), and 0.5 mM TCEP supplemented with 1 mM EDTA,. Cells were lysed by passing them three times through the Emulsiflex C5 homogenizer at 15,000 psi and the lysate was clarified by centrifugation in the Ti-45 rotor at 40,000 rpm for 35 min. The lysate was bound to 17 ml of glutathione sepharose (GE Healthcare, Uppsala, Sweden) by stirring for 1 hr at 4°C. Beads were harvested by centrifugation and washed with

10 column volumes of 137 mM NaCl, 2.7 mM KCl, 10 mM Na$_2$HPO$_4$, and 1.8 mM KH$_2$PO$_4$ (PBS buffer) supplemented with 1 mM EDTA and 0.5 mM TCEP. The beads were then re-suspended in 25 ml of PBS buffer containing 1 mM EDTA and 0.5 mM TCEP and incubated with 400 units of thrombin (Haematologic Technologies, Essex Junction, Vermont, USA) overnight at 4°C to cleave complexin from GST. The supernatant containing released complexin was concentrated to 4 ml using a 3000 molecular weight cutoff dialysis cassette (Millipore, Billerica, MA) and PMSF was added to 1 mM to quench the thrombin activity. The sample was then injected on a Superdex 200 (16/60) column that was equilibrated in 20 mM HEPES, pH 7.5, 100 mM NaCl, 4 mM DTT. Fractions containing complexin protein were combined and concentrated using a 3000 molecular weight cutoff dialysis cassette (Millipore, Billerica, MA). The protein concentration was measured by absorbance at 280 nm and aliquots were flash frozen in liquid nitrogen.

## Protein reconstitution

We used the same membrane compositions and protein densities as in our previous studies (*Diao et al., 2012*; *Kyoung et al., 2013*). Likewise, the reconstitution protocol was similar (*Kyoung et al., 2013*) with several changes: the concentration of 2-mercaptoethanol in the dialysis and vesicle-free buffers was changed from 1% to 0.1%, an extra dialysis step was added to ensure removal of Ca$^{2+}$ and detergent, all dialysis buffers contained 0.8 g/l Chelex 100 resin (Bio-Rad, Hercules, CA) and 2.5 g/l Bio-beads SM2 (Bio-Rad, Hercules, CA), and the Sepharose CL-4B column (GE Healthcare, Uppsala, Sweden) was packed under constant pressure.

Briefly, SV vesicles were reconstituted with both synaptotagmin-1 and synaptobrevin-2 (except for specified conditions in *Figure 4* were SV vesicles were only reconstituted with synaptobrevin-2), while PM vesicles were reconstituted with syntaxin-1A and SNAP-25A, using the previously described lipid compositions. The protein-to-lipid ratios used were 1:200 for synaptobrevin-2 and syntaxin-1A, and 1:1000 for synaptotagmin-1. A 3–5-fold excess of SNAP-25 (with respect to syntaxin-1A) and 3.5 mol% PIP$_2$ were added to the protein-lipid mixture for PM vesicles only. Dried lipid films were dissolved in 110 mM OG buffer containing purified proteins. Detergent-free buffer (20 mM HEPES, pH 7.4, 90 mM NaCl, 0.1% 2-mercaptoethanol) was then added to the protein-lipid mixture until the detergent concentration reached the critical micelle concentration of 24.4 mM. The vesicles were then subjected to size exclusion chromatography using a Sepharose CL-4B column, packed under near constant pressure by gravity with a peristaltic pump (GE Healthcare, Uppsala, Sweden) in a 5 ml column with a 2 ml bed volume, that was equilibrated with *buffer V* (20 mM HEPES, pH 7.4, 90 mM NaCl, 20 µM EGTA, 0.1% 2-mercaptoethanol) followed by dialysis into 2 l of detergent-free buffer V supplemented with 5 g of Bio-beads SM2 and 0.8 g/l Chelex 100 resin. After 4 hr, the buffer was changed with 2 l of fresh buffer V containing Bio-beads and Chelex, and dialysis continued for 12 hr. During the preparation of SV vesicles, 50 mM sulforhodamine B (Invitrogen, Carlsbad, CA) was present in all solutions prior to the size exclusion chromatography step. As described previously (*Kyoung et al., 2011*), the presence and purity of reconstituted proteins was confirmed by SDS-PAGE of the vesicle preparations, and the directionality of the membrane proteins (facing outward) was assessed by chymotrypsin digestion followed by SDS-PAGE. The size distributions of the SV and PM vesicles were analyzed by cryo-EM, as described previously (*Diao et al., 2012*).

## Single vesicle–vesicle content-mixing assay
### Overview
SV vesicles were labeled with a soluble fluorescent content dye at a moderately self-quenching concentration; for simplicity in this work we did not include a lipid-dye since we were exclusively interested in the exchange of content, the correlation for neurotransmitter release. The PM vesicles were immobilized on a surface that was passivated with polyethylene glycol (PEG) and functionalized via streptavidin-biotin linkages. SV vesicles were then added in the absence or presence of complexin-1 at the specified concentration. At variance with our previous work (*Diao et al., 2012*; *Kyoung et al., 2013*) where we incubated the SV vesicles for 20 min, we directly started monitoring the arrival of SV vesicles to surface-immobilized PM vesicles during the first minute acquisition period (*Figure 1C*, we refer to this as the vesicle association period—we prefer not to use the term 'docking' since it has a different meaning during the lifecycle of a synaptic vesicle [*Südhof, 2013*]). A stepwise increase in fluorescence emission of a spot in the field of view indicated the formation of a SV-PM vesicle pair during the vesicle association period.

Unbound SV vesicles were then removed through extensive washing with vesicle-free buffer, while continuing real-time observation of the fluorescence intensity (*Figure 1C*); consequently we did not observe any additional SV-PM vesicle associations after the washing step. While continuing the observation for another one-minute period, a second step-wise increase of fluorescence intensity appeared for some fraction of the associated SV vesicles, which indicated $Ca^{2+}$-independent, that is, spontaneous fusion events (referred to has the spontaneous fusion period). Next, we injected 500 µM $Ca^{2+}$ solution, and continued monitoring for another minute, referred to as the $Ca^{2+}$-triggered fusion period. For associated SV vesicles that did not undergo spontaneous fusion during the second period, a step-wise increase in fluorescence intensity during the third period indicated a $Ca^{2+}$ triggered fusion event. To determine the temporal arrival of $Ca^{2+}$ in the evanescent field of our TIR microscope setup, soluble Cy5 dye was added with the $Ca^{2+}$ buffer to monitor the emergence of fluorescence intensity (*Figure 1C*, bottom panel). Thus, our improved single vesicle–vesicle assay enables one to monitor the association of SV vesicles, spontaneous, and $Ca^{2+}$ triggered fusion events during the same data acquisition. Our system is thus ideally suited to study the opposing functions of complexin on SV-PM vesicle association, spontaneous release, and $Ca^{2+}$-triggered release.

## Details

Surface passivation of quartz slides was performed by coating the surface with polyethylene glycol (PEG) molecules to eliminate non-specific binding of vesicles. The same protocol and quality controls (surface coverage and non-specific binding) were used as described previously (*Diao et al., 2013a*; *Kyoung et al., 2013*), except that PEG-SVA (Laysan Bio, Inc.) instead of mPEG-SCM (Laysan Bio, Inc.) was used since it has a longer half life. The surface was functionalized by inclusion of biotin-PEG during pegylation. A quartz slide was assembled into a flow chamber and incubated with neutravidin (0.1 mg/ml). Biotinylated PM vesicles (100× dilution) were immobilized by incubation at room temperature (~25°C) for 30 min, followed by three rounds of washing with 120 µl buffer V, in order to remove unbound PM vesicles; each buffer wash effectively replaces the (~3 µl) flow chamber volume >100 times. Consequently, we did not observe any additional SV-PM vesicle association events during the subsequent observation periods. Upon the start of illumination and recording of the fluorescence from a particular field of view of the flow chamber, SV vesicles (diluted 100 to 1000 times) were loaded into the flow chamber to directly monitor vesicle association of SV vesicles to PM vesicles for one minute; when complexin-1 was included in a particular experiment, it was added concurrently together with the SV vesicles (*Figure 1C*). While continuing the recording, the flow chamber was washed three times with 120 µl of buffer V (including complexin-1 at the specified concentration, if applicable) in order to remove unbound SV vesicles. Subsequently, we continued recording for another minute to monitor spontaneous fusion events. To initiate $Ca^{2+}$-triggered fusion events within the same field of view, a solution consisting of buffer V, 500 µM $Ca^{2+}$, 500 nM Cy5 dye molecules (used as an indicator for the arrival of $Ca^{2+}$ in the evanescent field), and, if applicable, complexin-1 was injected into the flow chamber. The injection was performed at a speed of 66 µl/s by a motorized syringe pump (Harvard Apparatus, Holliston, MA) using a withdrawal method similar to the one described previously (*Kyoung et al., 2013*).

In our previous work (*Diao et al., 2012*; *Kyoung et al., 2013*), we used each flow chamber to monitor just one round of fusion events, which limited the throughput of individual studies and wasted most of the sample in the flow chamber. In order to increase the throughput of the assay and make better use of the vesicle samples, after intensive washing (3 × 120 µl) with buffer V containing 20 µM EGTA to remove $Ca^{2+}$ from the sample chamber, we repeated the entire acquisition procedure (vesicle association, spontaneous fusion, and $Ca^{2+}$-triggered fusion) in a different imaging area within the same flow chamber. This process was repeated for a total of five rounds. SV vesicles were diluted 1000× for the first and second repeat rounds, 200× for the third and fourth repeat rounds, and 100× for the fifth repeat round in order to offset the slightly increasing saturation of the surface with SV vesicles.

All experiments were performed on a prism-type total internal reflection fluorescence (TIRF) microscope using 532 nm (green) laser (CrystaLaser, Reno, NV) excitation. Two observation channels were created by a 640 nm single-edge dichroic beamsplitter (FF640-FDi01-25x36, Shemrock, Rochester, NY): one channel was used for the fluorescence emission intensity of the content dyes and the other one for that of the Cy5 dyes that are part of the injected $Ca^{2+}$-solution. The two channels were recorded on two adjacent rectangular areas (45 × 90 µm²) of a charged-coupled device (CCD) camera (iXon+ DV 897E, Andor Technology USA, South Windsor CT). The imaging data were recorded with the smCamera software from Taekjip Ha's group at the University of Illinois. Our procedure resulted in

a time series of images over a total of three minutes, consisting of the subsequent 1-min periods of vesicle association, spontaneous, and $Ca^{2+}$-triggered fusion, plus 5 s intervals for buffer exchanges. The arrival time of $Ca^{2+}$ was determined by monitoring of the Cy5 channel.

Fluorescent spots in the content dye channel were detected, but only those spots were analyzed that showed exactly one stepwise increase during the immediately preceding vesicle association period; this procedure excludes SV vesicles that were already associated during a previous round when performing repeat experiments using the same flow chamber. Stepwise increases in the fluorescence intensity time traces (examples shown in *Figure 1C*) were automatically detected by the computer program described previously (*Diao et al., 2013c*; *Kyoung et al., 2013*) and manually checked to ensure correct performance of the automated procedure. Histograms for spontaneous and triggered fusion event occurrence were plotted with a time bin of 1 s and fitted to a single exponential decay function. We did not observe any instances of SV-PM vesicle associations that showed more than one stepwise increase of the content dye fluorescence intensity during the observation periods. In other words, at most one fusion event was observed per associated SV-PM vesicle pair, consistent with the notion that each diffraction limited spot corresponds to a single pair of SV-PM vesicles rather than multiple SV vesicles docked to one PM vesicle.

## Acknowledgements

We thank Brandon Choi for discussions and critical reading of the manuscript, Yeon-Kyun Shin for expression constructs of synaptotagmin-1, syntaxin-1A, and synaptorevin-2, Haekjip Ha for the smCamera program, and the National Institutes of Health for support (R37-MH63105 to ATB).

## Additional information

### Competing interests

ATB: Reviewing editor, *eLife*. The other authors declare that no competing interests exist.

### Funding

| Funder | Grant reference number | Author |
| --- | --- | --- |
| Howard Hughes Medical Institute | | Axel T Brunger |
| National Institutes of Health | R37-MH63105 | Axel T Brunger |

The funders had no role in study design, data collection and interpretation, or the decision to submit the work for publication.

### Author contributions

YL, JD, Conception and design, Acquisition of data, Analysis and interpretation of data, Drafting or revising the article; DJC, YZ, Analysis and interpretation of data, Drafting or revising the article; RAP, MSP, Acquisition of data, Drafting or revising the article; ATB, Conception and design, Analysis and interpretation of data, Drafting or revising the article

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
