## [Decision Letter]

Thank you for sending your work entitled “Complexin Inhibits Spontaneous Release and Synchronizes Ca^2+^-triggered Synaptic Vesicle Fusion by Different Mechanisms” for consideration at *eLife*. Your article has been favorably evaluated by Randy Schekman (Senior editor) and 3 reviewers, one of whom is a member of our Board of Reviewing Editors.

The following individuals responsible for the peer review of your submission have agreed to reveal their identity: Richard Aldrich (Reviewing editor); James Rothman, and Yeon-Kyun Shin (peer reviewers).

The Reviewing editor and the other reviewers discussed their comments before we reached this decision, and the Reviewing editor has assembled the following comments to help you prepare a revised submission.

In this short report, Brunger and colleagues have elegantly extended their single vesicle content mixing assay to monitor vesicle recruitment (docking) and spontaneous fusion along calcium triggered fusion events. The work is technically difficult, and we have nothing but admiration for their success in doing so in this in vitro system. Using a soluble self-quenched fluorescent dye inside a vesicle that mimics the synaptic vesicles, they continuously monitor the recruitment of the vesicles (first 1 min acquisition), the spontaneous, calcium-independent fusion events (second 1 min acquisition) and the calcium-triggered fusion events triggered by injection of 500 µM calcium (third 1 min acquisition). They find that complexin increases the number of vesicles associated, suppresses spontaneous release and synchronizes calcium-triggered release. Mutations/truncation studies allow them to identify the role of different domains and these results in good agreement with previous studies done in cultured neurons. Lastly, they examine the effect of complexin concentration and report a different titration behavior for spontaneous release as compared to calcium-triggered release. Based on this they conclude that the complexin inhibits spontaneous release and calcium-triggered release via different mechanism.

What is new and interesting here is that the authors were able to observe the inhibition of content mixing in the absence of Ca^2+^. One other interesting observation is that the requirement of the very N-terminal region for the acceleration of vesicle fusion, which was previously observed by Sudhof and coworkers in cultured neurons. The results are the nice follow-up of the authors' previous work on complexin 1 and should be interesting to broad readership of *eLife*.

The following issues must be adequately addressed in a revised manuscript.

1) The notion that complexin inhibits spontaneous and calcium triggered release by different mechanisms is entirely rooted in the purportedly different concentration dependencies of Figure 3. This difference, if it exists, is rather small. The mid-points of the two curves are at about the same concentration of complexin, given the errors involved (indicated by the error bars). The authors should strongly tone down (including in the title) the conclusion of the paper especially since they also report that the same small domain/s in complexin are involved in suppressing spontaneous release and synchronizing Ca^2+^ release.

2) Significant technical point: The first 1 min of acquisition period is assigned as the 'vesicle association period' and the number of vesicles recruited shown in Figures 2 and 3 correspond to the 1 min time point. Are all docking events completed within that 1 min period? Or does docking continue beyond this time point even though un-docked vesicles are removed with a buffer wash? This is critical since the data shown in both Figures 2 and 3 are normalized to the number of associated vesicles. The spontaneous and calcium-triggered events are recorded in in the same observation cascade (the former after ∼1 min and later after ∼2 min) and if docking continues beyond the marked association period, then it could significantly alter the normalized data shown in Figures 2 and 3.

3) For the detection of spontaneous fusion the authors counted content mixed vesicle pairs during the first 1 min. We recommend that they run the experiment for several more minutes to increase the credibility of the data. Also, the experiment should be let to go all the way through to the end to see if the cumulative results actually saturate

4) Several years back Chapman and coworkers have proposed the possibility that synaptotagmin 1, instead of complexin, has the capacity of clamping SNARE-dependent membrane fusion. In this regards, the authors must test if this is the case with their in vitro experiments. The authors must carry out the control experiments without synaptotagmin 1 to examine the effect of complexin 1 on SNAREs in the absence synaptotagmin 1.

5) Effects of complexin 1 and one of its mutant on vesicle content mixing were previously examined by the same first author published in PNAS (2013). It is surprising that the PNAS paper by the same first author was not referenced. This paper must be referenced and discussed.

---

## [Author Response]

*1) The notion that complexin inhibits spontaneous and calcium triggered release by different mechanisms is entirely rooted in the purportedly different concentration dependencies of*
Figure 3*. This difference, if it exists, is rather small. The mid-points of the two curves are at about the same concentration of complexin, given the errors involved (indicated by the error bars). The authors should strongly tone down (including in the title) the conclusion of the paper especially since they also report that the same small domain/s in complexin are involved in suppressing spontaneous release and synchronizing Ca*^*2+*^
*release*.

We thank the reviewers for pointing out a weakness in our original description of the observed data. In response to their concerns we performed statistical significance tests of the titrations shows in Figure 3. We found that the Ca^2+^-triggered synchronization and amplitude exhibits a statistically significant increase between 50 and 100 nM complexin. In contrast, there is a statistically significant decrease of spontaneous fusion between 100 and 500 nM complexin. Thus, the difference in titration behaviors for spontaneous and Ca^2+^-triggered fusion is significant with a roughly ten-fold concentration difference in the most sensitive complexin concentration. Moreover, Figure 2 shows that the N-terminus, but not the C-terminus is essential for complexin’s effect on synchronization, but the reverse is true for complexin’s suppression of spontaneous fusion. Thus, we still feel that our data collectively suggest differences in mechanisms for these two functions. Moreover, genetic manipulations of the single complexin homolog in *Drosophila* show that the suppression and synchronization functions are genetically separable, also supporting the notion of distinct molecular mechanisms (PNAS 111, 10317-10322, 2014). Nevertheless we have softened and clarified the statements related to this point throughout the paper.

*2) Significant technical point: The first 1 min of acquisition period is assigned as the 'vesicle association period' and the number of vesicles recruited shown in*
Figures 2 and 3
*correspond to the 1 min time point. Are all docking events completed within that 1 min period? Or does docking continue beyond this time point even though un-docked vesicles are removed with a buffer wash? This is critical since the data shown in both*
Figures 2 and 3
*are normalized to the number of associated vesicles. The spontaneous and calcium-triggered events are recorded in in the same observation cascade (the former after ∼1 min and later after ∼2 min) and if docking continues beyond the marked association period, then it could significantly alter the normalized data shown in*
Figures 2 and 3.

We thank the reviewer for this comment that has been prompted by a rather terse description of the methods that we used. First, to avoid confusion we have changed the term “vesicle buffer” to “buffer V”. Please note that this is buffer only, with no vesicles. Thus, after the 1 minute observation time we wash out any free vesicles. Our washing procedure was as follows: while continuing the recording the flow chamber (∼ 3 μL) was washed three times with 120 μL of buffer V (including complexin at the specified concentration, if applicable) in order to remove unbound SV vesicles; thus each wash exchanges the sample volume roughly a hundred times. Consequently, we did not observe any additional docking events during the subsequent fusion observation periods. The normalization in Figure 2 and 3B-D used the number of events during the association period. Moreover, only those single vesicle-vesicle fluorescence intensity traces that exhibited a distinct event during the association period were analyzed for fusion events during the subsequent observation periods. We have added these explanations to the Materials and Methods section and the main text.

*3) For the detection of spontaneous fusion the authors counted content mixed vesicle pairs during the first 1 min. We recommend that they run the experiment for several more minutes to increase the credibility of the data. Also, the experiment should be let to go all the way through to the end to see if the cumulative results actually saturate*.

Unfortunately, photobleaching prevented us from continuing monitoring fluorescence intensity past ∼1 minute period. Instead, we performed multiple repeat experiments, leading to statistically significant differences in spontaneous fusion probabilities (Figures 2 and 3).

*4) Several years back Chapman and coworkers have proposed the possibility that synaptotagmin 1, instead of complexin, has the capacity of clamping SNARE-dependent membrane fusion. In this regards, the authors must test if this is the case with their in vitro experiments. The authors must carry out the control experiments without synaptotagmin 1 to examine the effect of complexin 1 on SNAREs in the absence synaptotagmin 1*.

We thank the reviewers for the suggestion and performed experiments with and without synaptotagmin as well as with and without wildtype complexin (new Figure 4 and associated discussion in the text). We do not find evidence for a clamping role of synaptotagmin in our system.

*5) Effects of complexin 1 and one of its mutant on vesicle content mixing were previously examined by the same first author published in PNAS (2013). It is surprising that the PNAS paper by the same first author was not referenced. This paper must be referenced and discussed*.

We thank the reviewer for the suggestion and now mention this paper in the concluding paragraphs.